# Threshold Dynamics of a Non-Linear Stochastic Viral Model with Time Delay and CTL Responsiveness

**DOI:** 10.3390/life11080766

**Published:** 2021-07-29

**Authors:** Jianguo Sun, Miaomiao Gao, Daqing Jiang

**Affiliations:** 1School of Science, China University of Petroleum (East China), Qingdao 266580, China; gaomm1991@upc.edu.cn (M.G.); daqingjiang2010@upc.edu.cn (D.J.); 2Nonlinear Analysis and Applied Mathematics (NAAM) Research Group, Department of Mathematics, Faculty of Science, King Abdulaziz University, Jeddah 121589, Saudi Arabia

**Keywords:** stochastic viral model, time delayed, CTL responsiveness, ergodic stationary distribution, extinction

## Abstract

This article focuses on a stochastic viral model with distributed delay and CTL responsiveness. It is shown that the viral disease will be extinct if the stochastic reproductive ratio is less than one. However, when the stochastic reproductive ratio is more than one, the viral infection system consists of an ergodic stationary distribution. Furthermore, we obtain the existence and uniqueness of the global positive solution by constructing a suitable Lyapunov function. Finally, we illustrate our results by numerical simulation.

## 1. Introduction

It is confirmed that approximately 100–250 million people are infected every year by different viruses, especially in regions of Asia and Africa [1]. To control epidemic viral diseases, an epidemic viral model is very useful, which can provide insights into the dynamics of viruses in vivo and offer a better understanding of viral diseases [2,3,4,5,6,7,8,9,10].

CTLs (Cytotoxic T Lymphocytes) play a significant role in antiviral mechanisms. On the one hand, CTLs imply the main immune factor inhibiting cell that limits the development of virus replication in vivo and depends on viral load [11,12,13]; on the other hand, it has recently been demonstrated that infected cells are killed not by the virus but by specific CTLs in some infectious diseases such as hepatitis B [3,14]. Therefore, the dynamics of the epidemic viral model with CTL responsiveness have drawn much attention from researchers in related areas [11,12,13,14,15,16].

The CTL immune response against a single pool of infected viral cells has been considered in [15,16,17], which is described by the following system:(1)$$\left\{\begin{array}{c} dx(t)=(\lambda -hx-exy)dt,\\ dy(t)=(exy-ay-pyz)dt,\\ dz(t)=(f(y,z)-bz)dt, \end{array}\right.$$
where the definitions of the variables are described in the following Table 1.

Note that time-series data of the immune state of patients look rather irregular. The possibility of dynamics of infinite delay has been introduced into the equations used in mathematical biology models since Volterra [18,19,20,21,22] translated the cumulative effect of the history of a system. K. Wang et al. [13] incorporate a time delay of the immune response in the system (Equation 1) to obtain the following system:(2)$$\left\{\begin{array}{c} dx(t)=(\lambda -hx(t)-ex(t)y(t))dt,\\ dy(t)=(ex(t)y(t)-ay(t)-py(t)z(t))dt,\\ dz(t)=(cy(t-\tau )-bz(t))dt, \end{array}\right.$$

They described the relationship between virus replication and the instantaneous immune response.

In the real world, epidemic models are also affected by escapable environmental white noise [5,6,7,8,9,23,24,25,26,27,28], full of randomness and stochasticity. As we understand it, there is little to no work researching the extinction and the ergodic stationary distribution of a system (Equation 2) with stochasticity and distributed delay, which is mainly due to the fact that Equation (Equation 2) is of a degenerate type. In this paper, we use an asymptotic approach [6,8,9,25] and give the following system:(3)$$\left\{\begin{array}{c} dx\left(t\right)=\left(\lambda -hx\left(t\right)-ex\left(t\right)y\left(t\right)\right)dt-\sigma_{1}xdB_{1}\left(t\right),\\ dy\left(t\right)=\left(ex\left(t\right)y\left(t\right)-ay\left(t\right)-py\left(t\right)z\left(t\right)\right)dt-\sigma_{2}ydB_{2}\left(t\right),\\ dz\left(t\right)=\left(cw\left(t\right)-bz\left(t\right)\right)dt-\sigma_{3}zdB_{3}\left(t\right),\\ dw(t)=\sigma (y(t)-w(t))dt, \end{array}\right.$$

The main purpose of this paper is to study the ergodic stationary distribution and extinction of the system (Equation 3). The existence and uniqueness of the global positive solution are also introduced.

The remainder of the paper is organized as follows. In Section 2, we introduce some necessary results throughout this paper. In Section 3, we show the uniqueness and positivity of global positive solutions of a stochastic system (Equation 3) with any positive initial value. In Section 4, we prove the existence and uniqueness of an ergodic stationary distribution of the solutions to the system (Equation 3) by constructing a suitable stochastic Lyapunov function; furthermore, we establish the persistence in the mean of the solutions of the system (Equation 3). In Section 5, we establish sufficient conditions for the extinction of the viral model. Some numerical simulations are introduced to demonstrate the theoretical results and reveal the effects of white noise. Finally, some concluding remarks and future directions are presented to close this paper.

## 2. Preliminaries

Throughout this paper, we first give some basic conceptions, as in [6]. Let (Ω,F,${\left\{F_{t}\right\}}_{t\geq 0},P)$ be a complete probability space with a filtration ${\left\{F_{t}\right\}}_{t\geq 0}$ satisfying the usual conditions (i.e., it is right continuous, and $F_{0}$ contains all P−null sets). Define $R_{+}^{4}\;=\;\left\{x\in R^{4}\;:\;x_{i}>0\;for\;all\;1\leq i\leq 4\right\}$, ${\bar{R}}_{+}^{4}\;=\;\left\{x\in R^{4}\;:\;x_{i}\geq 0\;for\;all\;1\leq i\leq 4\right\}$. In addition, if g(t) is an integral function on t∈[0,∞), define $g^{\mu }=sup\left\{g\left(t\right)|t\geq 0\right\},$$g^{l}=inf\left\{g\left(t\right)|t\geq 0\right\}$.

Firstly, we consider the general 4-dimensional stochastic differential equation
(4)$$\begin{array}{c} dx\left(t\right)=l\left(x\left(t\right),t\right)dt+q\left(x\left(t\right),t\right)dB\left(t\right),\;\;\mathrm{for}\;\;t\geq t_{0} \end{array}$$
with initial value $x\left(t_{0}\right)=x_{0}\in R^{4}$, where B(t) denotes 4-dimensional standard Brownian motion defined on the above probability space. Define the differential operator *L* associated with Equation (Equation 4) by Mao [6] as
$$L=\frac{\partial }{\partial t}+\Sigma l_{i}\left(x,t\right)\frac{\partial }{\partial x_{i}}+\frac{1}{2}\Sigma {\left[q^{T}\left(x,t\right)q\left(x,t\right)\right]}_{ij}\frac{\partial^{2}}{\partial x_{i}\partial x_{j}}.$$

If *L* acts on a function $V\in C^{2,1}\left(R^{4}\times {\bar{R}}_{+};{\bar{R}}_{+}\right)$, then
$$LV\left(x,t\right)=V_{t}\left(x,t\right)+V_{x}\left(x,t\right)+\frac{1}{2}trac\left[q^{T}\left(x,t\right)V_{xx}\left(x,t\right)q\left(x,t\right)\right],$$
where $V_{t}=\frac{\partial V}{\partial t},V_{x}=\left(\frac{\partial V}{\partial x_{1}},\ldots ,\frac{\partial V}{\partial x_{4}}\right)$ and $V_{xx}={\left(\frac{\partial^{2}V}{\partial x_{i}\partial x_{j}}\right)}_{4\times 4}$. Let x(t) be a homogeneous Markov process in $R^{4}$, which is described as the following stochastic differential equation by It${\hat{o}}^{\prime }$s formula [6]:$$dV\left(x\left(t\right),t\right)=LV\left(x\left(t\right),t\right)dt+V_{x}\left(x\left(t\right),t\right)r\left(x\left(t\right),t\right)dB\left(t\right).$$

The diffusion matrix is defined as follows:$$A\left(x\right)=\left(a_{ij}\left(x\right)\right),\;a_{ij}=\sum_{r=1}^{4}g_{r}^{i}\left(x\right)g_{r}^{j}\left(x\right).$$

## 3. Existence and Uniqueness of the Global Positive Solution

To study the dynamical behaviors of a viral model, where the solution is global and positive, because the coefficients of the system (Equation 3) do not satisfy the linear growth, the solutions of the system (Equation 3) may explode at a finite time. In this section, we show that there is a unique global positive solution of the system (Equation 3) from the idea in [8], the main theorem as follows.

**Theorem** **1.**
*For any initial value $\left(x\left(0\right),y\left(0\right),z\left(0\right),w\left(0\right)\right)\in R_{+}^{4}$, there is a unique positive solution (x(t),y(t),z(t),w(t)) of system (Equation 3) on t≥0, and the solution will remain in $R_{+}^{4}$ with a probability of one; that is to say, $\left(x\left(t\right),y\left(t\right),z\left(t\right),w\left(t\right)\right)\in R_{+}^{4}$ for all t≥0, almost surely.*


**Proof.** Our proof is based on the works of Mao et al. [9]. We know the coefficients of system (Equation 3) are locally Lipschitz continuous; thus, there is a unique solution (x(t),y(t),z(t),w(t)) on $\left[0,\tau_{0}\right)$ for any initial value $\left(x\left(0\right),y\left(0\right),z\left(0\right),w\left(0\right)\right)\in R_{+}^{4}$, where $\tau_{0}$ is an explosion time. If $\tau_{0}=\infty$ a.s, we can determine that the local solution is global. Let $n_{0}$ be a sufficiently large positive number for every component of (x(0),y(0),z(0),w(0)) lying in $\left[\frac{1}{n_{0}},n_{0}\right]$; for any $n\geq n_{0}$, the stopping time
(5)$$\begin{array}{c} \tau_{n}=\inf\left\{t\in \left[0,\tau_{0}\right):\min\left\{x\left(t\right),y\left(t\right),z\left(t\right),w\left(t\right)\right\}\leq \frac{1}{n}or\max\left\{x\left(t\right),y\left(t\right),z\left(t\right),w\left(t\right)\right\}\geq n\right\}. \end{array}$$We set infϕ=∞. Obviously, $\tau_{n}$ is increasing as n→∞. Set $\tau_{\infty }=\lim_{n\to \infty }\tau_{n}$; thus, $\tau_{\infty }\leq \tau_{0}$ a.s. If we can prove that $\tau_{\infty }=\infty$ a. s., then $\tau_{0}=\infty$, which implies $\left(x\left(t\right),y\left(t\right),z\left(t\right),w\left(t\right)\right)\in R_{+}^{4}$ a. s. for all t≥0. If $\tau_{\infty }<\infty$ a.s., it is the same. If we want to complete the proof, we should verify that $\tau_{\infty }=\infty$ a. s. If this assertion is false, there are two constants T≥0 and ε∈(0,1), such that
$$P\{\tau_{\infty }\leq T\}\geq \varepsilon .$$In addition, there is an integer $n_{1}\geq n_{0}$, such that
$$P\left\{\tau_{n}\leq T\right\}\geq \varepsilon \;for\;all\;n\geq n_{1}.$$We define a fundamental $C^{2}$-function $U:R_{+}^{4}\to {\bar{R}}_{+}$, which is
(6)$$\begin{array}{c} U=\left(x\left(t\right)-\alpha -\alpha \ln\frac{x(t)}{\alpha }\right)+\left(y\left(t\right)-\beta -\beta \ln\frac{y(t)}{\beta }\right)+\left(z\left(t\right)-1-\ln z\left(t\right)\right)+\gamma \left(w\left(t\right)-1-\ln w\left(t\right)\right), \end{array}$$
where α,β,γ are positive constants, which will be determined in the following text. The non-negativity of the function *U* can be seen from x−1−lnx≥0 for any x>0.Applying It${\hat{o}}^{\prime }$s formula [6], we obtain
(7)$$\begin{array}{c} dU\left(x,y,z,w\right)=LUdt-\sigma_{1}\left(x\left(t\right)-\alpha \right)dB_{1}\left(t\right)-\sigma_{2}\left(y\left(t\right)-\beta \right)dB_{2}\left(t\right)-\sigma_{3}\left(z\left(t\right)-1\right)dB_{3}\left(t\right), \end{array}$$
where
(8)$$\begin{array}{cc} LU(x,y,z,w) & =\left(1-\frac{\alpha }{x}\right)dx+\left(1-\frac{\beta }{y}\right)dy+\left(1-\frac{1}{z}\right)dz+\gamma \left(1-\frac{1}{w}\right)dw+\frac{1}{2}(\alpha \sigma_{1}^{2}+\beta \sigma_{2}^{2}+\sigma_{3}^{2}\\ & =\lambda -dx-exy-\frac{\alpha \lambda }{x}+d\alpha +\alpha ey+exy-ay-pyz-\beta e\;x\\ & +\beta a+\beta pz+cw-bz-\frac{cw}{z}+b+\gamma \sigma y-\gamma \sigma w-\frac{\gamma \sigma y}{w}+\gamma \sigma +\frac{1}{2}(\alpha \sigma_{1}^{2}+\beta \sigma_{2}^{2}+\sigma_{3}^{2}. \end{array}$$Choosing
$$\alpha =\frac{a-c}{e},\beta =\frac{b}{p},\gamma =\frac{c}{\sigma },$$
such that αe+γσ=a,βp=b,γσ=c, we can then obtain
(9)$$\begin{array}{ccc} LU\left(x,y,z,w\right)\leq \lambda +ad+\beta a+b+\gamma \sigma \leq \lambda +ad+\frac{ab}{p}+b+c+\frac{1}{2}\left(\frac{a-c}{e}\sigma_{1}^{2}+\frac{b}{p}\sigma_{2}^{2}+\sigma_{3}^{2}\right)\;\;\;\; & := & \;\;\;\;K, \end{array}$$
where *K* is a positive constant. The remainder of the proof is similar to Theorem 3.1 in Mao [9]. Hence, we omit it here. □

## 4. Ergodic Stationary Distribution

Here, we present some theories about the stationary distribution in this section. Although there is no endemic equilibrium point of the stochastic system (Equation 3), we want to obtain the existence of an ergodic stationary distribution, which indicates the persistence of the disease. Firstly, we define $R_{0}^{*}$ as a stochastic reproductive ratio of the system (Equation 3), such as
$$R_{0}^{*}=\frac{\lambda ea}{{\left(a+\frac{1}{2}\sigma_{2}^{2}\right)}^{2}\left(d+\frac{1}{2}\sigma_{1}^{2}\right)},$$
which is equal to $R_{0}=\frac{\lambda e}{ad}$ when $\sigma_{1}=\sigma_{2}=0$ [13]. Some known results about the theory of Has’Minskii are found in [28].

**Lemma** **1**([28])**.**
*The Markov process X(t) has a unique ergodic stationary distribution μ(·) if there exists a bounded domain $U\subset E_{l}$ with regular boundary *Γ*, and*
(*A*.1)*there is a positive number M such that $\sum_{i,j=1}^{l}a_{ij}\left(x\right)\xi_{i}\xi_{j}\geq M{\left|\xi \right|}^{2},\;x\in U,\;\xi \in R^{l}.$*(*A*.2)*there exists a non-negative $C^{2}$ function V, such that LV is negative for any $E_{l}\backslash U$. Then,*$$P_{x}\left\{\lim_{T\to \infty }\frac{1}{T}\int_{0}^{T}f\left(X\left(t\right)\right)dt=\int_{E_{l}}f\left(x\right)\mu \left(dx\right)\right\}=1,$$*for all $x\in E_{l}$, where f(·) is a function integrable with respect to the measure μ.*

Based on the theory of Has’minskii [28], we will give conditions which guarantee the existence of an ergodic stationary distribution.

**Theorem** **2.**
*Assume that $R_{0}^{*}>1$, then the solution (x(t),y(t),z(t),w(t)) of system (Equation 3) has an ergodic unique stationary distribution for any initial value $\left(x\left(0\right),y\left(0\right),z\left(0\right),w\left(0\right)\right)\in R_{+}^{4}.$*


**Proof.** The proof of Theorem 2 should satisfy the conditions of Lemma 1. Verify that (A.1) holds. Apparently, the corresponding diffusion matrix of system (Equation 3) is given by
$$A=\left(\begin{array}{cccc} \sigma_{1}^{2}x^{2} & 0 & 0 & 0\\ 0 & \sigma_{2}^{2}y^{2} & 0 & 0\\ 0 & 0 & \sigma_{3}^{2}z^{2} & 0\\ 0 & 0 & 0 & w^{2} \end{array}\right).$$Choosing $\tilde{M}=\min_{\left(x,y,z,w\right)\in D_{\varepsilon }}\left\{\sigma_{1}^{2}x^{2},\sigma_{2}^{2}y^{2},\sigma_{3}^{2}z^{2},w^{2}\right\}>0$, we obtain
$$\sum_{i,j=1}^{4}a_{ij}\left(x,y,z,w\right)\xi_{i}\xi_{j}=\sigma_{1}^{2}x^{2}\xi_{1}^{2}+\sigma_{2}^{2}y^{2}\xi_{2}^{2}+\sigma_{3}^{2}z^{2}\xi_{3}^{2}+w^{2}\xi_{4}^{2}\geq \tilde{M}|\xi {|}^{2},$$
for all $\left(x,y,z,w\right)\in D_{\varepsilon },\xi =\left(\xi_{1},\xi_{2},\xi_{3},\xi_{4}\right)\in R_{+}^{4}$, where
$$D_{\varepsilon }=\left\{\left(x,y,z,w\right)\in R_{+}^{4}:\varepsilon <x<\frac{1}{\varepsilon },\varepsilon <y<\frac{1}{\varepsilon },\varepsilon <z<\frac{1}{\varepsilon },\varepsilon <w<\frac{1}{\varepsilon }\right\}$$
is a bounded closed set, and ε>0 is a sufficiently small number. Thus, condition (A.1) is completed.Now, we construct a $C^{2}$-function $V:R_{+}^{4}\to R$ as follows:
$$V\left(x,y,z,w\right)=M\left(-\ln x-c_{0}\ln y\right)-c_{1}\ln y-\ln z-\ln w+\frac{1}{\theta +1}{\left(x+y+\frac{a}{4c}z+\frac{a}{2\sigma }w\right)}^{\theta +1},$$
where $\theta \in (0,min\frac{a}{a+\sigma_{2}^{2}},\frac{b}{b+\sigma_{3}^{2}},\frac{d}{d+\sigma_{1}^{2}})$ is a positive constant, and $c_{1}\leq -\frac{2+b+\sigma +\frac{1}{2}\left(\sigma_{2}^{2}+\sigma_{3}^{2}\right)}{a}.$When $R_{0}^{*}>1$, the constant *M* satisfies the following condition:
$$0<M\leq \frac{-2}{\left(d+\frac{1}{2}\sigma_{1}^{2}\right)\left(1-R_{0}^{*}\right)}.$$Applying It${\hat{o}}^{\prime }$s formula to the function V(x,y,z,w), denote
$$V_{1}=-\ln x-c_{0}\ln y,\;\;\;V_{2}=-c_{1}\ln y,$$
$$V_{3}=-\ln z,\;\;\;\;\;\;V_{4}=-\ln w,$$
$$V_{5}=\frac{1}{\theta +1}{\left(x+y+\frac{a}{4c}z+\frac{a}{2\sigma }w\right)}^{\theta +1}.$$We can apply the differential operator *L* to the above functions, respectively,
(10)$$\begin{array}{cc} LV_{1} & =-\frac{1}{x}x^{\prime }-c_{0}\frac{1}{y}y^{\prime }+\frac{1}{2}\left(\sigma_{1}^{2}+c_{0}\sigma_{2}^{2}\right)\\ & =-\frac{\lambda }{x}+d+ey-c_{0}ex+c_{0}a+c_{0}pz+\frac{1}{2}\left(\sigma_{1}^{2}+c_{0}\sigma_{2}^{2}\right)\\ & \leq -2\sqrt{\lambda c_{0}e}+c_{0}a+\frac{1}{2}c_{0}\sigma_{2}^{2}+d+\frac{1}{2}\sigma_{1}^{2}+ey+c_{0}pz. \end{array}$$Supposing
$$g\left(c_{0}\right)=-2\sqrt{\lambda c_{0}e}+c_{0}a+\frac{1}{2}c_{0}\sigma_{2}^{2},$$
$$g^{\prime }\left(c_{0}\right)=-\frac{\lambda e}{\sqrt{\lambda c_{0}e}}+a+\frac{1}{2}\sigma_{2}^{2}=0,$$
we can obtain
$$c_{0}=\frac{\lambda e}{{\left(a+\frac{1}{2}\sigma_{2}^{2}\right)}^{2}}.$$Hence,
(11)$$\begin{array}{cc} LV_{1} & =g\left(c_{0}\right)+d+\frac{1}{2}\sigma_{1}^{2}+ey+c_{0}p\;z\\ & \leq -\frac{\lambda ea}{{\left(a+\frac{1}{2}\sigma_{2}^{2}\right)}^{2}}+d+\frac{1}{2}\sigma_{1}^{2}+ey+c_{0}p\;z\\ & =\left(d+\frac{1}{2}\sigma_{1}^{2}\right)\left(1-\frac{\lambda ea}{{\left(a+\frac{1}{2}\sigma_{2}^{2}\right)}^{2}\left(d+\frac{1}{2}\sigma_{1}^{2}\right)}\right)+ey+\frac{\lambda ep}{{\left(a+\frac{1}{2}\sigma_{2}^{2}\right)}^{2}}z\\ & =\left(d+\frac{1}{2}\sigma_{1}^{2}\right)\left(1-R_{0}^{*}\right)+ey+\frac{\lambda ep}{{\left(a+\frac{1}{2}\sigma_{2}^{2}\right)}^{2}}z; \end{array}$$
(12)$$LV_{2}=-c_{1}\frac{1}{y}y^{\prime }+\frac{1}{2}\sigma_{1}^{2}=-c_{1}ex+c_{1}a+c_{1}pz+\frac{1}{2}\sigma_{2}^{2};$$
(13)$$LV_{3}=-\frac{1}{z}z^{\prime }+\frac{1}{2}\sigma_{3}^{2}=-\frac{cw}{z}+b+\frac{1}{2}\sigma_{3}^{2};$$
(14)$$LV_{4}=-\frac{1}{w}w^{\prime }=-\frac{\sigma y}{w}+b+\sigma ;$$
(15)$$\begin{array}{cc} LV_{5} & ={\left(x+y+\frac{a}{4c}z+\frac{a}{2\sigma }w\right)}^{\theta }\left(x^{\prime }+y^{\prime }+\frac{a}{4c}z^{\prime }+\frac{a}{2\sigma }w^{\prime }\right)\\ & +\frac{\theta }{2}{\left(x+y+\frac{a}{4c}z+\frac{a}{2\sigma }w\right)}^{\theta -1}\left(\sigma_{1}^{2}x^{2}+\sigma_{2}^{2}y^{2}+{\left(\frac{a}{4c}\right)}^{2}\sigma_{3}^{2}z^{2}\right)\\ & \leq {\left(x+y+\frac{a}{4c}z+\frac{a}{2\sigma }w\right)}^{\theta }\left(\lambda -dx-\frac{a}{2}y-\frac{a}{4}w-\frac{ab}{4c}z\right)\\ & +\frac{\theta }{2}{\left(x+y+\frac{a}{4c}z+\frac{a}{2\sigma }w\right)}^{\theta -1}\left(\sigma_{1}^{2}x^{2}+\sigma_{2}^{2}y^{2}+{\left(\frac{a}{4c}\right)}^{2}\sigma_{3}^{2}z^{2}\right)\\ & \leq \lambda {\left(x+y+\frac{a}{4c}z+\frac{a}{2\sigma }w\right)}^{\theta }-\left(dx+\frac{a}{2}y+\frac{ab}{4c}z+\frac{a}{4}w\right)\left(x^{\theta }+y^{\theta }+{\left(\frac{a}{4c}\right)}^{\theta }z^{\theta }+{\left(\frac{a}{2\sigma }\right)}^{\theta }w^{\theta }\right)\\ & +\frac{\theta }{2}{\left(x+y+\frac{a}{4c}z+\frac{a}{2\sigma }w\right)}^{\theta -1}\left(\sigma_{1}^{2}x^{2}+\sigma_{2}^{2}y^{2}+{\left(\frac{a}{4c}\right)}^{2}\sigma_{3}^{2}z^{2}\right)\\ & \leq \lambda {\left(x+y+\frac{a}{4c}z+\frac{a}{2\sigma }w\right)}^{\theta }-dx^{\theta +1}-\frac{a}{2}y^{\theta +1}-{\left(\frac{a}{4c}\right)}^{\theta +1}bz^{\theta +1}-\frac{a}{4}{\left(\frac{a}{2\sigma }\right)}^{\theta }w^{\theta +1}\\ & +\frac{\theta }{2}{\left(x+y+\frac{a}{4c}z+\frac{a}{2\sigma }w\right)}^{\theta -1}\left(\sigma_{1}^{2}x^{2}+\sigma_{2}^{2}y^{2}+{\left(\frac{a}{4c}\right)}^{2}\sigma_{3}^{2}z^{2}\right)\\ & \leq B-d\theta x^{\theta +1}-\frac{a}{2}\theta y^{\theta +1}-{\left(\frac{a}{4c}\right)}^{\theta +1}b\theta z^{\theta +1}-\frac{a^{\theta +1}}{2^{\theta +3}\sigma^{\theta }}w^{\theta +1}, \end{array}$$
where
$$B=sup\{\lambda {\left(x+y+\frac{a}{4c}z+\frac{a}{2\sigma }w\right)}^{\theta }+\frac{\theta }{2}{\left(x+y+\frac{a}{4c}z+\frac{a}{2\sigma }w\right)}^{\theta -1}\left(\sigma_{1}^{2}x^{2}+\sigma_{2}^{2}y^{2}+{\left(\frac{a}{4c}\right)}^{2}\sigma_{3}^{2}z^{2}\right),\;-d\left(1-\theta \right)x^{\theta +1}-\frac{a}{2}\left(1-\theta \right)y^{\theta +1}-{\left(\frac{a}{4c}\right)}^{\theta +1}b\left(1-\theta \right)z^{\theta +1}-\frac{a^{\theta +1}}{2^{\theta +4}\sigma^{\theta }}w^{\theta +1}\},$$
where $\theta \in (0,min\frac{a}{a+\sigma_{2}^{2}},\frac{b}{b+\sigma_{3}^{2}},\frac{d}{d+\sigma_{1}^{2}})$ is a positive constant, and B<0. From the above analysis, we have
(16)$$\begin{array}{cc} LV(x,y,z,w) & =MLV_{1}+LV_{2}+LV_{3}+LV_{4}+LV_{5}\\ & \leq M(\left(d+\frac{1}{2}\sigma_{1}^{2}\right)\left(1-\frac{\lambda ea}{{\left(a+\frac{1}{2}\sigma_{2}^{2}\right)}^{2}\left(d+\frac{1}{2}\sigma_{1}^{2}\right)}\right)+ey+\frac{\lambda ep}{{\left(a+\frac{1}{2}\sigma_{2}^{2}\right)}^{2}}z)\\ & -c_{1}ex+c_{1}a+c_{1}pz+\frac{1}{2}\sigma_{2}^{2}\\ & -\frac{1}{z}z^{\prime }+\frac{1}{2}\sigma_{3}^{2}=-\frac{cw}{z}+b+\frac{1}{2}\sigma_{3}^{2}-\frac{1}{w}w^{\prime }=-\frac{\sigma y}{w}+b+\sigma \\ & +B-d\theta x^{\theta +1}-\frac{a}{2}\theta y^{\theta +1}-{\left(\frac{a}{4c}\right)}^{\theta +1}b\theta z^{\theta +1}-\frac{a^{\theta +1}}{2^{\theta +3}\sigma^{\theta }}w^{\theta +1}, \end{array}$$
and we define
$$f_{1}\left(x\right)=c_{1}a+b+\sigma +\frac{1}{2}\left(\sigma_{2}^{2}+\sigma_{3}^{2}\right)-c_{1}ex-d\theta x^{\theta +1},$$
$$f_{2}\left(y\right)=M\left(\left(d+\frac{1}{2}\sigma_{1}^{2}\right)\left(1-\frac{\lambda ea}{{\left(a+\frac{1}{2}\sigma_{2}^{2}\right)}^{2}\left(d+\frac{1}{2}\sigma_{1}^{2}\right)}\right)+ey\right)-\frac{a}{2}\theta y^{\theta +1},$$
$$f_{3}\left(z\right)=M\frac{\lambda ep}{{\left(a+\frac{1}{2}\sigma_{2}^{2}\right)}^{2}}z+c_{1}pz-\frac{cw}{z}-{\left(\frac{a}{4c}\right)}^{\theta +1}b\theta z^{\theta +1},$$
$$f_{3}\left(w\right)=B-\frac{\sigma y}{w}-\frac{a^{\theta +1}}{2^{\theta +3}\sigma^{\theta }}w^{\theta +1}.$$We can divide $R_{+}^{4}\backslash D_{\varepsilon }$ into the following eight domains:
$$D_{1}=\left\{\left(x,y,z,w\right)\in R_{+}^{4}:0<x<\varepsilon \right\};\;\;\;\;\;D_{2}=\left\{\left(x,y,z,w\right)\in R_{+}^{4}:0<y<\varepsilon \right\};$$
$$D_{3}=\left\{\left(x,y,z,w\right)\in R_{+}^{4}:0<z<\varepsilon ,\varepsilon <w<\frac{1}{\varepsilon }\right\};\;\;\;\;\;D_{4}=\left\{\left(x,y,z,w\right)\in R_{+}^{4}:\varepsilon <z<\frac{1}{\varepsilon },0<w<\varepsilon \right\};$$
$$D_{5}=\left\{\left(x,y,z,w\right)\in R_{+}^{4}:x>\frac{1}{\varepsilon }\right\};\;\;\;\;\;D_{6}=\left\{\left(x,y,z,w\right)\in R_{+}^{4}:y>\frac{1}{\varepsilon }\right\};$$
$$D_{7}=\left\{\left(x,y,z,w\right)\in R_{+}^{4}:z>\frac{1}{\varepsilon }\right\};\;\;\;\;\;D_{8}=\left\{\left(x,y,z,w\right)\in R_{+}^{4}:w>\frac{1}{\varepsilon }\right\}.$$Clearly, $D_{\varepsilon }={⋃}_{j=1}^{8}D_{j}$. In the following text, we will show that LV(x,y,z,w)≤−1 on $R_{+}^{4}\backslash D_{\varepsilon }$, which is equivalent to prove it on the above eight domains.Case 1. If $\left(x,y,z,w\right)\in D_{1}$, one can choose
$$c_{1}\leq -\frac{2+b+\sigma +\frac{1}{2}\left(\sigma_{2}^{2}+\sigma_{3}^{2}\right)}{a},$$
and
$$LV\left(x,y,z,w\right)\leq c_{1}a+b+\sigma +\frac{1}{2}\left(\sigma_{2}^{2}+\sigma_{3}^{2}\right)-c_{1}ex-d\theta x^{\theta +1}\leq -2;$$Case 2. If $\left(x,y,z,w\right)\in D_{2}$, one can choose $R_{0}^{*}>1$ and
$$0<M\leq \frac{-2}{\left(d+\frac{1}{2}\sigma_{1}^{2}\right)\left(1-R_{0}^{*}\right)},$$
$$LV\left(x,y,z,w\right)\leq M\left(\left(d+\frac{1}{2}\sigma_{1}^{2}\right)\left(1-R_{0}^{*}\right)+ey\right)-\frac{a}{2}\theta y^{\theta +1}\leq -2;$$Case 3. If $\left(x,y,z,w\right)\in D_{3}$,
$$LV\left(x,y,z,w\right)\leq M\frac{\lambda ep}{{\left(a+\frac{1}{2}\sigma_{2}^{2}\right)}^{2}}z+c_{1}pz-\frac{cw}{z}-{\left(\frac{a}{4c}\right)}^{\theta +1}b\theta z^{\theta +1}\leq -\frac{cw}{z}\leq -2;$$Case 4. If $\left(x,y,z,w\right)\in D_{4}$,
$$LV\left(x,y,z,w\right)\leq B-\frac{\sigma y}{w}-\frac{a^{\theta +1}}{2^{\theta +3}\sigma^{\theta }}w^{\theta +1}\leq -\frac{\sigma y}{w}\leq -2;$$Case 5. If $\left(x,y,z,w\right)\in D_{5}$,
$$LV\left(x,y,z,w\right)\leq c_{1}a+b+\sigma +\frac{1}{2}\left(\sigma_{2}^{2}+\sigma_{3}^{2}\right)-c_{1}ex-d\theta x^{\theta +1}\leq -c_{1}ex-d\theta x^{\theta +1}\leq -2;$$Case 6. If $\left(x,y,z,w\right)\in D_{6}$,
(17)$$\begin{array}{cc} LV(x,y,z,w) & \leq M\left(\left(d+\frac{1}{2}\sigma_{1}^{2}\right)\left(1-R_{0}^{*}\right)+ey\right)-\frac{a}{2}\theta y^{\theta +1}\\ & \leq M\left(\left(d+\frac{1}{2}\sigma_{1}^{2}\right)\left(1-R_{0}^{*}\right)+ey\right)-\frac{a}{2}\theta {\left(\frac{1}{\varepsilon }\right)}^{\theta +1}\\ & \leq -2; \end{array}$$Case 7. If $\left(x,y,z,w\right)\in D_{7}$, because $M\frac{\lambda ep}{{\left(a+\frac{1}{2}\sigma_{2}^{2}\right)}^{2}}+c_{1}p-{\left(\frac{a}{4c}\right)}^{\theta +1}b\theta {\left(\frac{1}{\varepsilon }\right)}^{\theta }<0$, we determine that
(18)$$\begin{array}{cc} LV(x,y,z,w) & \leq M\frac{\lambda ep}{{\left(a+\frac{1}{2}\sigma_{2}^{2}\right)}^{2}}z+c_{1}pz-\frac{cw}{z}-{\left(\frac{a}{4c}\right)}^{\theta +1}b\theta z^{\theta +1}\\ & \leq z\left(M\frac{\lambda ep}{{\left(a+\frac{1}{2}\sigma_{2}^{2}\right)}^{2}}+c_{1}p\right)-cw\varepsilon -{\left(\frac{a}{4c}\right)}^{\theta +1}b\theta z^{\theta +1}\\ & \leq z(M\frac{\lambda ep}{{\left(a+\frac{1}{2}\sigma_{2}^{2}\right)}^{2}}+c_{1}p-{\left(\frac{a}{4c}\right)}^{\theta +1}b\theta {\left(\frac{1}{\varepsilon }\right)}^{\theta })\\ & \leq -\frac{1}{\varepsilon }\\ & \leq -2; \end{array}$$Case 8. If $\left(x,y,z,w\right)\in D_{8}$, because B<0, we determine that
$$LV\left(x,y,z,w\right)\leq B-\frac{\sigma y}{w}-\frac{a^{\theta +1}}{2^{\theta +3}\sigma^{\theta }}w^{\theta +1}\leq B-\frac{a^{\theta +1}}{2^{\theta +3}\sigma^{\theta }}{\left(\frac{1}{\varepsilon }\right)}^{\theta +1}\leq -2.$$Therefore, for all $\left(x,y,z,w\right)\in R_{+}^{4}\backslash D_{\varepsilon }$, V(x,y,z,w)≤−1, which indicates that assumption (A.2) holds.We can know that the system (Equation 3) is ergodic and has a unique stationary distribution. This completes the proof. □

## 5. Extinction of the System 3

For the dynamical behavior of epidemic viral models, the main concern is finding the condition in which the virus will be eradicated in a long time when $R_{0}^{*}>1$. In this section, we shall consider the extinction of the system (Equation 3).

According to the results in [9], we can obtain the following lemma.

**Lemma** **2.**
*For any initial value, the solution of the stochastic model satisfies*
(19)$$\lim_{t\to \infty }\frac{\ln x(t)}{t}\leq 0,\;\lim_{t\to \infty }\frac{\ln y(t)}{t}\leq 0,\;\lim_{t\to \infty }\frac{\ln z(t)}{t}\leq 0,\;\lim_{t\to \infty }\frac{\ln w(t)}{t}\leq 0\;a.s.$$
(20)$$\lim_{t\to \infty }\frac{x(t)+y(t)+z(t)+w(t)}{t}=0,\;a.s..$$

*Moreover,*
(21)$$\lim_{t\to 0}\frac{1}{t}\int_{0}^{t}x\left(m\right)dB_{1}\left(m\right)=0\;,\lim_{t\to 0}\frac{1}{t}\int_{0}^{t}y\left(m\right)dB_{2}\left(m\right)=0,\lim_{t\to 0}\frac{1}{t}\int_{0}^{t}z\left(m\right)dB_{3}\left(m\right)=0\;\;a.s..$$


**Theorem** **3.**
*Let (x,y,z,w) be the solution of system (Equation 3) with any initial value $\left(x\left(0\right),y\left(0\right),z\left(0\right),w\left(0\right)\right)\in R_{+}^{4}$. If $R_{0}^{*}<1$, then the solution (x,y,z,w) of system (Equation 3) satisfies*
$$\underset{t\to \infty }{lim sup}\frac{\ln y(t)}{t}\leq \left(a+\frac{1}{2}\sigma_{2}^{2}\right)\left(R_{0}^{*}-1\right)<0\;\;a.s..$$

*Namely, the disease will be eradicated in the long term.*


**Proof.** Applying Itô’s formula to lny(t), we obtain
(22)$$\begin{array}{cc} d\ln y(t) & =\left(ex-a-pz-\frac{1}{2}\sigma_{2}^{2}\right)dt+\sigma_{2}dB_{2}\left(t\right)\\ & \leq \left[ex-\left(a+\frac{1}{2}\sigma_{2}^{2}\right)\right]dt+\sigma_{2}dB_{2}\left(t\right). \end{array}$$Integrating the above formula from 0 to *t* on both sides, then
$$\ln y\left(t\right)-\ln y\left(0\right)\leq \int_{0}^{t}\left[ex-\left(a+\frac{1}{2}\sigma_{2}^{2}\right)\right]ds+\int_{0}^{t}\sigma_{2}dB_{2}\left(s\right).$$According to the strong law of large numbers [29], we have
$$\lim_{t\to 0}\frac{1}{t}\int_{0}^{t}dB_{2}\left(s\right)=0\;\;\;\;a.s..$$For,
(23)$$\begin{array}{cc} d\ln x(t) & =\frac{1}{x}\left(\lambda -dx\left(t\right)-ex\left(t\right)y\left(t\right)-x\sigma_{1}^{2}\right)dt+\sigma_{1}dB_{1}\left(t\right)\\ & \leq \left[\frac{\lambda }{x}-\left(d+\frac{1}{2}\sigma_{1}^{2}\right)\right]dt+\sigma_{1}dB_{1}\left(t\right). \end{array}$$From (Equation 23), we can obtain
$$d\ln x\left(t\right)\geq -\left(d+\frac{1}{2}\sigma_{1}^{2}\right)dt,$$
and
(24)$$\begin{array}{c} \frac{1}{d\ln x(t)}\leq -\frac{1}{(d+\frac{1}{2}\sigma_{1}^{2})dt}. \end{array}$$Meanwhile,
$$\frac{\lambda }{x}\geq d\ln x\left(t\right)+\left(d+\frac{1}{2}\sigma_{1}^{2}\right)+\sigma_{1}dB_{1}\left(t\right),$$
which indicates that
(25)$$\begin{array}{c} x\leq \frac{\lambda }{d\ln x\left(t\right)+\left(d+\frac{1}{2}\sigma_{1}^{2}\right)+\sigma_{1}dB_{1}\left(t\right)}\leq \frac{\lambda }{d+\frac{1}{2}\sigma_{1}^{2}}+\frac{\lambda }{d\ln x\left(t\right)+\sigma_{1}dB_{1}\left(t\right)}. \end{array}$$There exists a small number $\sigma_{2}$, such that
(26)$$\begin{array}{cc} \int_{0}^{t}x\left(s\right)ds & \leq \frac{\lambda }{d+\frac{1}{2}\sigma_{1}^{2}}t+\int_{0}^{t}\frac{\lambda }{d\ln x\left(s\right)+\sigma_{1}dB_{1}\left(s\right)}ds\\ & \leq \frac{\lambda }{d+\frac{1}{2}\sigma_{1}^{2}}t+\lambda \int_{0}^{t}\frac{1}{d\ln x(s)}ds\\ & \leq \frac{\lambda }{d+\frac{1}{2}\sigma_{1}^{2}}t-\lambda \int_{0}^{t}\frac{1}{\left(d+\frac{1}{2}\sigma_{1}^{2}\right)}ds\\ & \leq \frac{\lambda }{d+\frac{1}{2}\sigma_{1}^{2}}t\\ & \leq \frac{\lambda a}{\left(d+\frac{1}{2}\sigma_{1}^{2}\right)\left(a+\frac{1}{2}\sigma_{2}^{2}\right)}t. \end{array}$$Taking the superior limit and using the stochastic comparison theorem, combining (Equation 26), we obtain
(27)$$\begin{array}{cc} \underset{t\to \infty }{lim sup}\frac{\ln y(t)}{t} & =\underset{t\to \infty }{lim sup}e\frac{1}{t}\int_{0}^{t}x\left(s\right)ds-\left(a+\frac{1}{2}\sigma_{2}^{2}\right)\\ & \leq e\frac{\lambda a}{\left(d+\frac{1}{2}\sigma_{1}^{2}\right)\left(a+\frac{1}{2}\sigma_{2}^{2}\right)}-\left(a+\frac{1}{2}\sigma_{2}^{2}\right)\\ & =\left(a+\frac{1}{2}\sigma_{2}^{2}\right)\left(\frac{\lambda ea}{{\left(a+\frac{1}{2}\sigma_{2}^{2}\right)}^{2}\left(d+\frac{1}{2}\sigma_{1}^{2}\right)}-1\right)\\ & =\left(a+\frac{1}{2}\sigma_{2}^{2}\right)\left(R_{0}^{*}-1\right)\\ & <0\;\;\;\;a.s.. \end{array}$$Therefore, this indicates that
$$\lim_{t\to \infty }y\left(t\right)=0\;\;\;\;a.\;s..$$Consequently, this means that the virus will be eradicated in a long time. This completes the proof. □

## 6. Examples and Numerical Simulations

In this section, we will introduce some examples and numerical simulations to demonstrate the above theoretical results. Using the Milstein higher-order method developed in [23], we obtain the discretization equation of the system (Equation 3).
(28)$$\left\{\begin{array}{c} x\left(k+1\right)=x\left(k\right)+\left[\lambda -dx\left(k\right)-ex\left(k\right)y\left(k\right)\right]▵t+\sigma_{1}x\left(k\right)\sqrt{▵t}\xi_{k}+\frac{\sigma_{1}^{2}x\left(k\right)}{2}▵t\left(\xi_{k}^{2}-1\right),\\ y\left(k+1\right)=y\left(k\right)+\left[ex\left(k\right)y\left(k\right)-ay\left(k\right)-py\left(k\right)z\left(k\right)\right]▵t+\sigma_{2}y\left(k\right)\sqrt{▵t}\eta_{k}+\frac{\sigma_{2}^{2}y\left(k\right)}{2}▵t\left(\eta_{k}^{2}-1\right),\\ z\left(k+1\right)=z\left(k\right)+\left[cw\left(k\right)-bz\left(k\right)\right]▵t+\sigma_{3}z\left(k\right)\sqrt{▵t}\zeta_{k}+\frac{\sigma_{3}^{2}z\left(k\right)}{2}▵t\left(\zeta_{k}^{2}-1\right),\\ w(k+1)=w(k)+\sigma [y(k)-w(k)]▵t. \end{array}\right.$$
where the time increment ▵t is positive, and $\xi_{k},\eta_{k},\zeta_{k}$ are the Gaussian random variables which follow the distribution N(0,1),k=1,2,3.

In system (Equation 3), according to [27]:

**Example** **1.**
*In order to check the existence of an ergodic stationary distribution, we choose the values of the system parameters as follows: $\left(\sigma_{1},\sigma_{2},\sigma_{3}\right)=\left(0.1,0.1,0.1\right)$, λ=1000,d=0.1,e=0.002,a=5,p=0.2,c=0.2,b=0.3,σ=0.2, then $R_{0}^{*}=3.208>1,$ where $R_{0}^{*}$ is defined before Theorem 2 and choosing $c_{1}=-15,M=7$. In other words, the conditions of Theorem 2 hold, and there is an ergodic stationary distribution μ(·) of system (Equation 3), which will persist for a long time. Figure 1 confirms this.*


**Example** **2.**
*In order to check the extinction of the system (Equation 3), we choose the values of the system parameters as follows: $\left(\sigma_{1},\sigma_{2},\sigma_{3}\right)=\left(0.1,0.1,0.1\right)$, λ=250,d=0.1,e=0.002,a=5,p=0.2,c=0.2,b=0.3,σ=0.2; then, $R_{0}^{*}=0.849<1,$ where $R_{0}^{*}$ is defined before Theorem 3 and choosing $c_{1}=-20,M=5$. In other words, the conditions of Theorem 3 hold, which will be extinct in a long time. Figure 2 confirms this.*


## 7. Discussion

This paper considers the parameters *d*, *a*, and *b* disturbed by the white noise and assumes the dynamics of the corresponding stochastic system (Equation 3) with the time delay and CTL responsiveness. The reason for choosing the three disturbed parameters is that these three parameters are important in controlling the viral disease. Of course, in the following research, we will focus on the general situation to investigate the influence of white noise.

## Figures and Tables

**Figure 1 life-11-00766-f001:**
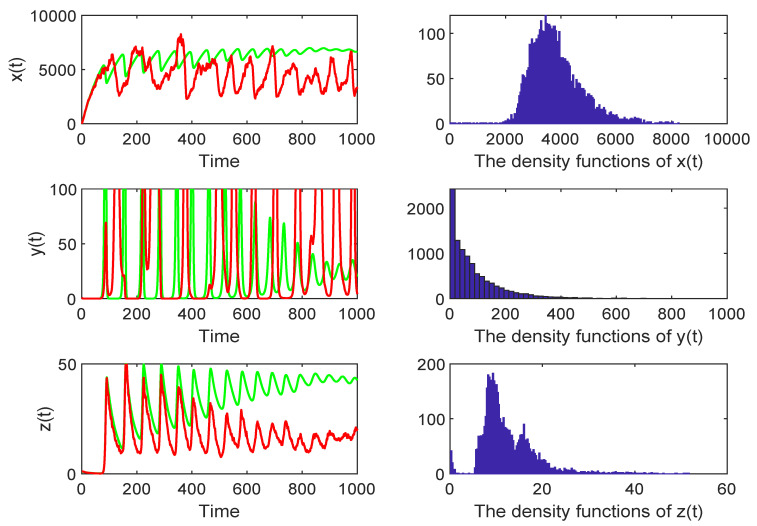
The red curve represents the result of the stochastic model, whereas the green curve represents the result of the deterministic model. The right column shows the density function of the stochastic model (Equation 3). Choosing $\left(\sigma_{1},\sigma_{2},\sigma_{3}\right)=\left(0.1,0.1,0.1\right)$, when $R_{0}^{*}=3.208>1$, the disease will persist for a long time.

**Figure 2 life-11-00766-f002:**
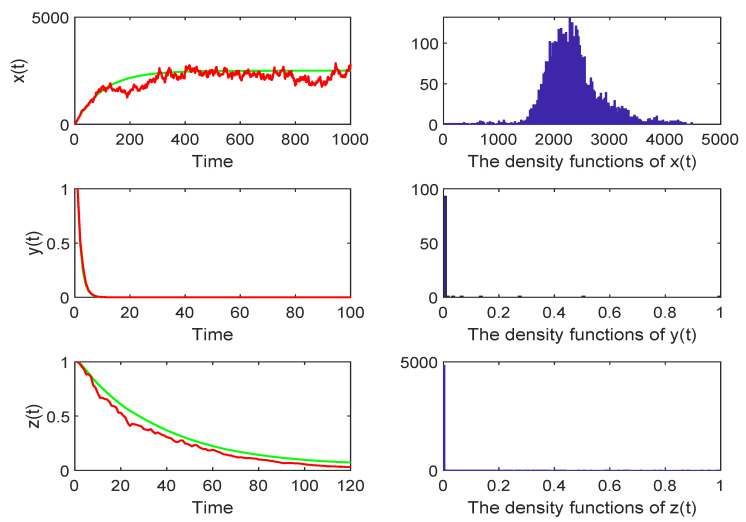
The red curve represents the result of the stochastic model, whereas the green curve represents the result of the deterministic model. The right column shows the density function of the stochastic model (Equation 3). Choosing $\left(\sigma_{1},\sigma_{2},\sigma_{3}\right)=\left(0.1,0.1,0.1\right)$, when $R_{0}^{*}=0.849<1$, the disease will be extinct in a long time.

**Table 1 life-11-00766-t001:** Variables in the model (adapted from [15,16]).

Variables	Definition
x(t)	the number of susceptible host cells at time *t*
y(t)	the virus population at time *t*
z(t)	the number of CTLs at time *t*
λ	susceptible host cells generation rate
hx	susceptible host cells death rate
exy	rate at which susceptible host cells become infected by the virus
ay	infected cells death rate
pyz	rate at which infected cells are killed by the CTL
f(y,z)	rate of immune response due to virus activation

## Data Availability

Not applicable.

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
