# Peer review of "Threshold Dynamics of a Non-Linear Stochastic Viral Model with Time Delay and CTL Responsiveness"

_life, 2021, doi:10.3390/life11080766_

Round 1

Reviewer 1 Report

The authors consider a stochastic version of an system of ODEs that describe the population dynamics of immune responses to persistent viruses.  This topic may be of interest both from a practical point of view as well as from a mathematical point of view. Unfortunately, the quality of the presentation is considerably sub-standard which makes it hard to judge the contribution of the paper. Therefore I cannot recommend its publication. Some detailed comments are listed below. 

  • The authors seem to have a distinct preference for using the symbol "d". This makes reading the paper very confusing. d is sometimes a constant, sometimes a dimension, sometimes the differential operator.
  • The introduction of the three Brownian terms need to be motivated. At present, it is not entirely clear how the SDE (1.3) relates to the original ODE (1.1) or to the DDE (1.2)
  • Be consistent in your notation f in section 1 and f in the preliminaries is not the same. You first introduce notation in d-dimensional space and then move to n-dimensional space. It seems you introduce preliminaries from different sources, but do not adapt the notation to the present setting.
  • Probably the preliminaries can be collected in an appendix
  • Try to avoid pages with only formulas. Most readers will prefer some guidance to figure out the formulas.

Reviewer 2 Report

This paper provides a better way to understand the dynamics of virus in vivo, especially the understanding of viral diseases. The authors gave the main attention to a stochastic viral model with distributed delay and CTL responsiveness, which is closed to the real world. Meanwhile, the authors gave the conditions of the extinction and the ergodic stationary distribution of the viral disease. The article has more practical application value. The article is well written and readable. I think the article meets the publishing requirements of the magazine (Life).

However, there are some minor problems that need to be modified and can be accepted after modification.

  • : The line 41 in page 3 should change the reason to reasons.
  • : There is an extra bracket in the equation 3.5 in page 4
  • : The punctuation should be unified in the equation 4.2
  • : A period is missed in the equation 5.8 in page 9
  • : The authors should give the value of R0* in the describe of the Fig1 and Fig 2
